# Tigecycline Pharmacokinetic and Pharmacodynamic Profile in Patients with Chronic Obstructive Pulmonary Disease Exacerbation

**DOI:** 10.3390/antibiotics12020307

**Published:** 2023-02-02

**Authors:** Maria Kipourou, Olga Begou, Katerina Manika, Georgios Ismailos, Paschalina Kontou, Georgia Pitsiou, Helen Gika, Ioannis Kioumis

**Affiliations:** 1Pulmonary Department, 424 General Military Hospital, 56429 Thessaloniki, Greece; 2Department of Chemistry, Aristotle University of Thessaloniki, 54124 Thessaloniki, Greece; 3Biomic AUTh, Center for Interdisciplinary Research and Innovation (CIRI-AUTH), 54124 Thessaloniki, Greece; 4Pulmonary Department, Aristotle University of Thessaloniki, G. Papanikolaou Hospital, 54629 Thessaloniki, Greece; 5Experimental-Research Center ELPEN, ELPEN Pharmaceuticals, Leoforos Marathonos 95, 19009 Pikermi, Greece; 61st Intensive Care Unit, G. Papanikolaou Hospital, 54629 Thessaloniki, Greece; 7Respiratory Failure Unit, Aristotle University of Thessaloniki, G. Papanikolaou Hospital, 54629 Thessaloniki, Greece; 8Laboratory of Forensic Medicine & Toxicology, School of Medicine, Aristotle University of Thessaloniki, 54224 Thessaloniki, Greece; 9Professor Emeritus, Aristotle University of Thessaloniki, 54124 Thessaloniki, Greece; 10Euromedica, Geniki Kliniki, 54645, Thessaloniki, Greece

**Keywords:** tigecycline, COPD, pharmacokinetics, pharmacodynamics

## Abstract

Background: We aimed to evaluate the pharmacokinetic profile of tigecycline in plasma and its penetration to sputum in moderately ill patients with an infectious acute exacerbation of chronic obstructive pulmonary disease (COPD). Methods: Eleven patients hospitalized with acute respiratory failure due to an acute COPD exacerbation with clinical evidence of an infectious cause received tigecycline 50 mg twice daily after an initial loading dose of 100 mg. Blood and sputum samples were collected at steady state after dose seven. Results: In plasma, mean C_max_ pl was 975.95 ± 490.36 ng/mL and mean C_min_ pl was 214.48 ±140.62 ng/mL. In sputum, mean C_max sp_ was 641.91 ± 253.07 ng/mL and mean C_min sp_ was 308.06 ± 61.7 ng/mL. In plasma, mean AUC _0–12 pl_ was 3765.89 ± 1862.23 ng*h/mL, while in sputum mean AUC _0–12 sp_ was 4023.27 ± 793.37 ng*h/mL. The mean penetration ratio for the 10/11 patients was 1.65 ± 1.35. The mean Free AUC_0–24 pl_/MIC ratio for *Streptococcus pneumoniae* and *Haemophilus influenzae* was 25.10 ± 12.42 and 6.02 ± 2.97, respectively. Conclusions: Our findings support the clinical effectiveness of tigecycline against commonly causative bacteria in COPD exacerbations and highlight its sufficient lung penetration in pulmonary infections of moderate severity.

## 1. Introduction

Tigecycline is a glycylcycline belonging to the newest Group 3 of the tetracyclines [1]. A unique substitution of a N-alkyl-glycylamido group at position 9 on the D-ring of minocycline is the chemical basis of tigecycline’s ability to overcome the two major tetracycline resistance mechanisms of ribosomal protection and active efflux [2]. Tigecycline has a broad spectrum of antimicrobial activity against Gram-positive pathogens, namely *Enterococcus faecalis*, *Staphylococcus aureus*, *Streptococcus pneumoniae*, *Streptococcus pyogenes*, as well as against Gram-negative ones, i.e., *Acinetobacter baumannii*, *Enterobacter cloacae*, *Escherichia coli*, *Haemophilus influenzae*, *Klebsiella pneumoniae*, and *Legionella pneumophilia* [3]. Furthermore, tigecycline is active against anaerobic bacteria (Bacteroides *fragilis, B. vulgatus*, and *B. uniformis*) and distinct rapid growing mycobacteria (*Mycobacterium abscessus*, and *fortuitum*), whereas, importantly, *Pseudomonas* spp., *Proteus* spp., *Providencia* spp., and *Morganella morganii* are intrinsically resistant to it [3,4]. Initial U.S. Food and Drug Administration (FDA) approval concerned complicated intra-abdominal, complicated skin and skin structure infections, while, in 2008, use for community acquired bacterial pneumonia was also approved [5]. In 2013, the FDA issued a boxed warning for tigecycline, underlining that is not an indicated treatment for hospital-acquired pneumonia (HAP) and ventilation-associated pneumonia (VAP) [6]. European Medicines Agency (EMA) suggests tigecycline use for intra-abdominal and complicated skin infections but not for infections for which other antimicrobials are more suitable [7]. Since then, tigecycline seems to seek its role in antimicrobial regimens for respiratory infections [8,9,10]. 

Chronic obstructive pulmonary disease (COPD) acute exacerbations are important events in the course of the disease that have a serious impact on disease morbidity and mortality [11,12]. Bacteria such as *Moraxella catarrhalis*, *Streptococcus pneumoniae*, *Haemophilus influenzae, Pseudomonas aeruginosa, Staphylococcus aureus*, and *Haemophilus parainfluenzae* are identified from sputum cultures in up to 50% of COPD exacerbations and viruses in 30–50% of them [11,13]. Anthonisen criteria are used in clinical practice, alone or in combination with inflammatory biomarkers, to diagnose those COPD exacerbations most likely triggered by bacteria and in need of antibiotic treatment [11,12]. The targeted use of antibiotics in COPD infectious exacerbations has favorable effects on recovery time, risk of early relapse, treatment failure, and hospitalization duration, according to the 2022 Global Initiative for Chronic Obstructive Lung Disease (GOLD) report [12].

Despite the beneficial effect of antibacterial agents in COPD acute exacerbations, there is a lack of evidence for antibiotic pharmacokinetics (PKs) and pharmacodynamics (PDs) in COPD patients [14]. COPD patients are more likely to be older, active or ex-smokers, underweight with low albumin levels, suffering from various comorbidities, and on various inhaled and systemic medication [15,16,17]. Apart from each one of these factors with a potential influence on antibiotic metabolism, COPD is, per se, a state of increased systemic inflammation, possibly constituting an additional factor affecting antibacterial agents’ PK/PD profile [12]. 

## 2. Materials and Methods

### 2.1. Study Design, Participants, and Procedures

This was a prospective open-label study designed to investigate the safety, adverse events, and pharmacokinetics of tigecycline administered in hospitalized patients with infectious and chronic obstructive pulmonary disease (COPD) acute exacerbation. The study was conducted in the Aristotle University of Thessaloniki Pulmonary Department in G. Papanikolaou Hospital and the Pulmonary Department of 424 General Military Hospital of Thessaloniki (424 GMHT), Greece, between June 2016 and December 2019. All patients provided written informed consent on study enrollment. The study was approved by the Aristotle University Medical School Bioethics Committee (No: 154/2015). 

Patients were eligible for the study after hospital admission due to acute infectious COPD exacerbation. Diagnosis was either based on COPD history or the physician’s clinical judgement in the Emergency Department. All patients had a history of inhaled therapy with Long-Acting Beta Agonists (LABAs), mainly in combination with Long-Acting Muscarinic Agonists (LAMAs) or Inhaled Corticosteroids (ICSs). Acute COPD exacerbation diagnosis was based on patients’ clinical presentation and physical examination. 

All patients received standard care treatment with inhaled LABAs, LAMAs, systematic corticosteroids, and oxygen therapy, depending on individual needs. Antibiotic use was based on Anthonisen criteria, which require an acute deterioration of at least 2 symptoms, such as breathlessness, sputum volume, and sputum purulence. All patients included in our study had a minimum of 2 Anthonisen criteria positives. 

Since tigecycline is not effective against *P. aeruginosa*, all factors predisposing COPD patients to *P. aeruginosa* infection or colonization were exclusion criteria for the study: FEV1 < 30% in spirometry, >3 exacerbations in the past year, COPD patients on Long-Term Oxygen Therapy (LTOT), concomitant cardiac disease, use of antibiotics in the past 3 months, as well as COPD under long-term per os corticosteroids. Patients with evidence of pneumonia or bronchiectasis from the chest X-ray were also excluded. Sputum cultures were obtained from all patients on admission. 

A loading dose of 100 mg tigecycline (dose 1) was administered intravenously (i.v.) to all patients followed by 50 mg tigecycline i.v. every 12 h. Every tigecycline infusion duration was 1 h. Blood and sputum samples were collected at steady state after tigecycline dose 7 (on day 4 or 5 of hospitalization). For this purpose, blood samples were collected through a peripheral catheter positioned remotely from the catheter used for the treatment infusion on specific time points, i.e., just before the infusion of tigecycline and 12 h from the previous dose (time 0), on completion of the 1-h tigecycline infusion (time 1), and then 2, 3, 4, 6, 9, and 12 h after the i.v. administration. Blood samples were centrifuged, and the supernatant plasma was stored at −20 °C. 

Sputum samples were also collected on time points 0, 1, 2, 3, 4, 6, 9, and 12 h following the tigecycline i.v. administration. Sputum collection was prone to each patient’s production rate and ability to expectorate. All collected samples were homogenized and emulsified using an ultrasonic processor for 1 min and then stored at −20 °C (VCX-130; Sonics and Materials, Inc., Newton, MA, USA). 

### 2.2. Methods

#### 2.2.1. Sample Preparation

Plasma and sputum samples were stored at −80 °C and allowed to thaw at room temperature prior to analysis. For plasma samples, 100 μL were placed in an Eppendorf tube, and 600 μL of ice-cold acetonitrile (ACN) (−20 °C), containing 1 μg/mL of Internal Standard (IS) daptomycin 0.5 μg/mL, was added. The sample was vigorously shaken for 1 min and centrifuged for 10 min at 10,000× *g* at 4 °C. The clear supernatant was injected into the analytical system. 

Sputum samples after thawing were placed into an ultrasonic water bath sonicator for 2 min. Next, 100 μL of the sputum sample was diluted with 100 μL H_2_O, and then 20 μL of IS and 500 μL of ethyl acetate (EA) were added. The sample was vortex-mixed for 1 min and centrifuged for 10 min at 10,000× *g*, 4 °C. Then, 400 μL were collected from the upper organic phase and evaporated to dryness under vacuum. The residue was reconstituted with 50 μL of H_2_O: ACN, 95:5 *v*/*v* + 0.1% formic acid and injected into the analytical system. 

#### 2.2.2. LC-MS/MS Analysis

All samples were analyzed based on an LC-MS/MS method developed for the purposes of the study and validated based on bioanalytical guidelines. For serum samples, intra-day accuracy and precision were between 86–110% and 1–8.1%, accordingly, while LOQ was found to be 3 ng/mL. For sputum samples, intra-day accuracy ranged from 91.2% to 120.0%, and precision from 0.1% to 7.8% while LOQ was 57.7 ng/mL.

Chromatography was performed on an Acquity BEH C18 column (150 × 2.1 mm i.d., 1.7 μm; Waters) at 50 °C with an Acquity BEH C18 VanGuard pre-column (5 mm × 2.1 mm i.d., 1.7 μm; Waters). Mobile phase consisted of A. H_2_O + 0.1% formic acid and B. ACN + 0.1% formic acid. After a 0.5 min isocratic step at 100% A–0% B a linear gradient was applied from 0% to 100% B over 1 min at a flow rate of 0.4 mL/min. 

For serum samples, detection was performed using an ACQUITY UPLC H-Class system, whereas sputum samples were analyzed using ExionLC™ System SCIEX Triple Quad 6500+. A multiple reaction monitoring (MRM) mode was performed under optimized detection parameters.

Estimated plasma and sputum concentrations were plotted versus time and the corresponding pharmacokinetic data were analyzed using a noncompartmental model (Win-Nonlin software, v.8.3)

Penetration ratio was calculated by dividing the AUC _0–12 sputum_ / AUC 0–12 plasma for each patient. AUC_0–24 plasma_ was calculated by doubling the AUC_0–12 plasma_ for each patient_._ Free AUC_0–24 plasma_ was calculated, assuming the unbound fraction of tigecycline to be 0.20 [18]. The PD index of interest for tigecycline is free AUC_0–24_/MIC and was calculated by dividing fAUC_0–24 plasma_ with the only available FDA MIC breakpoints for *S.pneumoniae* (MIC ≤ 0.06 mcg/mL) and *H.infuenzae* (MIC ≤ 0.25 mcg/mL) [19]. The European Committee on Antimicrobial Susceptibility Testing (EUCAST) has recently stated that there is insufficient evidence (IE) that *H. influenzae* and *S. pneumoniae* constitute a good therapeutical target for tigecycline [20].

## 3. Results

In total, 11 patients with a running diagnosis of infectious COPD exacerbation were included in the study. Demographic and clinical characteristics of the patients are presented in Table 1. Five female and six male patients aged 51–88 years old, with a Body Mass Index (BMI) ranging from 24.2 to 49.3, received tigecycline. Mean C-reactive protein (CRP) value was 6.25 ± 3.18 mg/dl and mean white blood cell (WBC) count was 8476 ± 3365 cells/mm^3^. 

All patients presented respiratory failure upon admission, being in need of supplementary oxygen delivery. All patients provided sputum samples with the exception of patient 8. Sputum cultures were negative for all patients except for patient 11, whose culture turned out positive for Candida tropicalis on day 8 of hospitalization. 

All 11 patients had normal renal and liver function upon hospital admission and no laboratory deterioration was observed until hospital discharge for any patient in the study. 

### 3.1. Pulmonary Function 

Spirometry was performed for all patients on hospital admission and on hospital discharge (Table 1). There was only one patient with FEV_1_ < 30% (patient 5 FEV_1_: 24%); however, he was included in the study because he had two consecutive sputum cultures that were negative for *P. aeruginosa* prior to hospitalization. Mean FEV_1_% for the 11 patients was 43.18 ± 11.45% upon admission and 62.52 ± 19.52% upon hospital discharge, showing a mean increase of 47.7% during hospital stay. Accordingly, mean FVC% was 57.09 ± 8.84% upon admission and 77 ± 15.60% on hospital discharge, showing a 37.7% increase. Overall, there was only one patient (patient 7) showing a 11% decrease in FEV_1_ during his hospital stay; however, he had clinically improved regarding dyspnea, sputum volume, and sputum purulence. 

### 3.2. Pharmacokinetic (PK) Results

The estimated main pharmacokinetic parameters are displayed in Table 2. The mean plasma and sputum concentrations over time are displayed in Figure 1.

In plasma, mean C_max pl_ was 975.95 ± 490.36 ng/mL, while mean C_min pl_ was 214.48 ± 140.62 ng/mL. In sputum, mean C_max sp_ was 641.91 ± 253.07 ng/mL, and mean C_min sp_ was 308.06 ± 61.7 ng/mL. The mean time to reach C_max_ in sputum was 2.9 ± 1.7 h for the 10 patients with available sputum samples. 

In plasma, mean AUC _0–12 pl_ was 3765.89 ± 1862.23 ng*h/mL, while, in sputum, mean AUC _0–12 sp_ was 4023.27 ± 793.37 ng*h/mL. The mean penetration ratio for the 10/11 patients was estimated to be 1.65 ± 1.35.

The mean elimination half time (T_1/2_) was 17.69 ± 10.92 h, and the mean plasma clearance was 17.16 ± 22.17 L/h. 

### 3.3. Pharmacodynamic (PD) Results

We calculated the PD index best correlating with tigecycline clinical efficacy, fAUC_0–24 pl_/MIC, for one Gram-positive (*Streptococcus pneumoniae*) and one Gram-negative (*Haemophilus influenzae*) bacteria, which are often identified as the cause of infectious COPD acute exacerbations. For both *S. pneumoniae* and *H. influenzae*, a unique FDA susceptibility breakpoint exists for tigecycline, with no intermediate or resistant breakpoints [19]. The mean fAUC_0–24 pl_/MIC ratio for *S. pneumoniae* and *H. influenzae* was 25.10 ± 12.42 and 6.02 ± 2.97, respectively. Regarding non-severely ill patients, Rubino et al. have identified a tigecycline AUC/MIC value ≥ 12.8 predictive of clinical efficacy in community acquired pneumonia (CAP) [18]. In our study, 3 out of 11 patients did not achieve this threshold of efficacy for *S. pneumoniae*. On the other hand, an AUC/MIC ≤ 100 value has been identified as a risk factor for progression to CAP in COPD patients with acute exacerbation in the presence of a positive *S. pneumoniae* culture [21]. Consequently, we could assume that tigecycline treatment would not protect some of our patients from progression to CAP in the hypothetical presence of *S. pneumoniae*. 

## 4. Discussion

To our knowledge, this is the only PK study for tigecycline conducted in a ward of COPD patients with acute infectious exacerbation. These PK findings are, with limited exceptions, in accordance with the existing data, and show significant variation among the patients studied; moreover, they support the clinical efficacy of tigecycline in chronic respiratory patients, in view of a substantial lung penetration ratio.

Tigecycline pharmacokinetics remain controversial, with many unanswered questions, mainly due to their problematic clinical efficacy in severe lower respiratory illness, such as VAP and HAP [22,23,24]. In view of the increased mortality and the microbiological failure [25], high dosage regimens of tigecycline have been proposed in order to overcome the probable pharmacokinetic basis of reduced efficacy. In our study, we used the standard tigecycline regimen for the antimicrobial treatment of patients with mild acute COPD exacerbation; off-label use of this broad-spectrum antibiotic was based on the microbiological coverage offered by tigecycline, matching the bacteria most commonly causing the infectious COPD exacerbations, with the exception of *P. aeruginosa*. Even though the sputum cultures obtained from our patients did not lead to the identification of any causative microorganism, this is not unusual in everyday clinical practice, since bacteria are successfully cultured in barely 30% of COPD exacerbation cases [13]. Furthermore, recent data concerning the COPD lung microbiome demonstrate a complex microbiological diversity shift pattern, with various bacterial species proliferating during the acute exacerbations, while the airway microbiome is relatively stable during stable COPD [26]. 

Regarding the PK results of this study, the estimated plasma C_max_ 975.95 ± 490.36 ng/mL value is comparable to the findings reported by Gotfried et al. [27] and Cai et al. [28]. Both studies have used the standard dose regimen of tigecycline, the former for healthy individuals and the latter in severe pneumonia patients. Conte et al., in their tigecycline PK study on healthy individuals, using the standard dose, reported a C_max_ 0.72 ± 0.24 μg/mL and a great interindividual variability of tigecycline trough concentrations, based on the enrolled individuals’ body weight [29]. In accordance with this remark, in this study, patient 1, with a Body Mass Index (BMI): 46 kg/m^2^, was the patient with the lowest trough and maximum plasma concentrations. On the other hand, patient 6 was the patient with the highest BMI (49.3 kg/m^2^), even though his plasma concentrations were not as low as anticipated compared to the other patients (Table 1 and Table 2).

Dimopoulos et al., using the high-dose tigecycline regimen for VAP patients, reported almost double plasma concentrations, as well as very high plasma AUC_0–12_ values, most likely due to the higher doses administered [30]. Plasma AUC_0–12_ values estimated in the current study were higher than the corresponding estimations of both Gotfried et al. [27] and Conte et al. [29] (3.7 ± 1.8 compared to 2.20 ± 0.42 and 1.73 ± 1.64 μg∙h/mL, respectively), even though their study population consisted of healthy individuals. On the other hand, Cai et al., using the standard tigecycline dose regimen for severe pneumonia ward patients, reported a considerably higher plasma AUC _0–12_ value of 9.13 ± 0.59 μg∙h/mL [28]. Consequently, it seems reasonable to assume that higher tigecycline regimens could lead to higher plasma concentrations. The apparent differences in plasma tigecycline levels between this study and those of Cai et al. (both studies using the standard tigecycline regimen in ward patients) could be attributed to ethnic and body size characteristics, as well as differences observed between patients [31]. 

There is limited data regarding tigecycline pulmonary concentrations. With the exception of this study, only Cai et al. [28] have reported sputum tigecycline concentrations and sputum AUC_0–12_ in a ward of pneumonia patients. Interestingly, in that PK study, all five patients were younger and with lower BMI values, and they presented sputum concentrations and sputum mean AUC_0–12_ values almost three-fold higher than the corresponding values obtained from our older and higher BMI patients. This marked difference could also be attributed to the lower degree of lung infection observed in the COPD patients of the present study compared to the CAP ones of Cai et al. [28]. 

Data from animal studies have clearly demonstrated the intensifying role of infection on the epithelial lining fluid (ELF) tigecycline penetration [32,33]. 

Furthermore, there is limited yet varied data regarding tigecycline concentration in ELF and alveolar cells (ACs), studied by means of bronchoscopy. ELF and AC antibiotic concentrations are of great value in cases of antibiotics used to treat lower respiratory tract infections, providing insights on drug penetration in the different lung compartments [34]. Tigecycline ELF and AC concentrations were first reported by Conte et al. in healthy individuals, receiving standard dose [29]. ELF C_max_ and C_min_ obtained by Conte et al. were similar to the results of Gotfried et al. [27] and De Pascale et al. [35], who administered the high-dose regimen. Dimopoulos et al. [30] recently reported two- to three-fold higher mean sputum AUC_0–12_ values compared to Conte et al. [29] (7.13 ± 2.61 μg∙h/mL compared to 2.28 μg∙h/mL, respectively), having used the high-dose tigecycline regimen on critically ill patients. On the other hand, Burkhart et al., in a PK study with three critically ill patients, reported extremely low steady state tigecycline ELF concentrations, despite high AC concentrations [36]. The estimated penetration ratio was extremely low (ELF/plasma ratio 0.03 ± 0.03 at 1 h and 0.18 ± 0.09 at 12 h from the infusion), raising concerns about tigecycline effectiveness against extracellular bacteria involved in the lower respiratory tract infections [36]. 

Interestingly, mean max and trough sputum concentrations in our study are analogous to the ELF concentrations reported by all three studies by De Pascale, Gotfried, and Conte [27,29,35]. Although the sputum collection has not been linked to optimal results in PK studies, due to a possible admixture of saliva resulting in the over-estimation of lung penetration, the concentrations measured in the current study are within the values of existing data, with the exception of the Cai et al. study [28]. On the other hand, although completely noninvasive as a procedure, sputum collection needs to overcome certain difficulties. Indeed, sputum collection on day 5 of antibiotic treatment was challenging since most of the patients had already experienced clinical improvement with a marked reduction in their sputum volume. In our experience, continuous encouragement of our COPD patients during sputum collection and direct visualization of the obtained sample can lead to good quality sputum specimens with minimal saliva admixture. 

Furthermore, the AC tigecycline concentrations in both studies with such available data [29,36] are manifold higher than ELF concentrations, highlighting tigecycline’s effectiveness against intracellular pathogens [34]. The penetration ratio in our study is in accordance with the median ELF to plasma concentration ratio of the Monte Carlo simulation by Rubino et al. [37], although a wide variation was noted (5th and 95th percentiles ratio for ELF penetration 0.561 and 5.230). Similarly, in the current study, the mean penetration ratio based on sputum samples showed a wide variation (1.65 ± 1.35). Additionally, tigecycline is known for its significant volume of distribution (Vd), suggestive of the advantageous tissue penetration in the intracellular compartments [4]. In our study, mean Vd was 151.95 ± 73.43 L, which is the lowest estimation compared to the corresponding ones of Conte et al. (634 ± 172 L) [29], De Pascale et al. (438.6 L) [35], and Gotfried et al. (315 ± 67 L) [27]. 

Clinical features (improvement of dyspnea, improvement in sputum volume and purulent content, as well as the reversal of the respiratory failure for all patients) and spirometry findings were suggestive of clinical improvement in our study. This could be anticipated by the moderate infection degree in our COPD exacerbating patients compared to severely ill patients receiving tigecycline. On the other hand, although the PD parameter of interest, mean AUC_0–24 pl_/MIC, was above the critical threshold suggested for CAP, it was well below this target in a minority of our patients (3/11). Absence of tigecycline intermediate and resistance breakpoints for *S. pneumoniae* have previously raised concerns of a misleading limited in vivo efficacy [38]. On the other hand, in a Monte Carlo simulation evaluating a standard tigecycline dosage regimen, tigecycline was found to be highly effective against Gram-positive strains but not against Gram-negative bacteria [39] while, in our study, PD results also support a better effect against *S. pneumoniae* compared to the Gram-negative *H. influenzae* (mean AUC_f 0–24 pl_/MIC: 25.10 ± 12.42 vs. 6.02 ± 2.97).

There are several limitations to our study. The COVID-19 pandemic led to major changes in clinical practices worldwide as well as in the pulmonary wards, where patients eligible for this study were hospitalized, which is the main cause for the limited sample size of our study. Furthermore, since this study was designed for nonintubated COPD patients in everyday clinical practice, sputum samples were collected instead of bronchoscopy lavages, which could provide tigecycline ELF concentrations.

## 5. Conclusions

This study adds to the limited available intrapulmonary PK data in a real-life ward setting of hospitalized COPD patients. The obtained results are subject to the limited number of patients studied and the sputum samples used to evaluate the tigecycline lung penetration; however, they are in accordance with most of the existing data. The tigecycline pharmacokinetic profile at a standard dosage regimen supports its clinical effectiveness against commonly causative bacteria in COPD exacerbations and highlights its sufficient lung penetration in pulmonary infections of moderate severity.

## Figures and Tables

**Figure 1 antibiotics-12-00307-f001:**
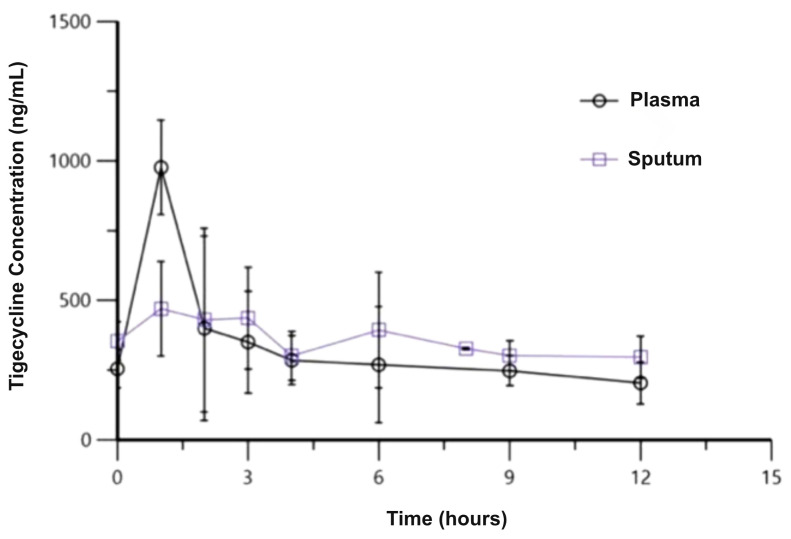
Tigecycline concentration (mean, ng/mL) versus time (hours) plots in plasma (circles) and sputum (squares) after the 7th administration of tigecycline standard dosage regimen to 11 patients hospitalized due to an acute COPD exacerbation.

**Table 1 antibiotics-12-00307-t001:** Demographic and clinical characteristics of the patients of the study treated with tigecycline.

	Gender (M: Male, F: Female)	Age (Years)	BMI (kg/m^2^)	FEV_1_ %on Admission	FVC% on Admission	Exacerbations/ Past Year	CRP (mg/dL) on Admission	WBC (Cells/mm^3^) on Admission
Patient 1	F	60	46.0	51	51	0	7.4	7400
Patient 2	M	73	32.8	28	51	0	3.5	7600
Patient 3	F	51	34.6	39	60	0	4.2	5700
Patient 4	F	77	37.6	54	66	1	3.6	7400
Patient 5	M	79	31.3	24	53	1	3.2	11,500
Patient 6	F	53	49.3	37	49	2	4.9	9440
Patient 7	M	82	30.8	44	68	1	13.5	16,700
Patient 8	M	60	37.0	59	59	0	6.0	11,100
Patient 9	M	88	30.0	61	71	0	6.9	6100
Patient 10	F	62	25.8	41	60	0	11.2	4300
Patient 11	M	74	24.2	37	40	1	4.5	6000
Mean ± SD	5F/6M	69 ± 12	34.4 ± 7.4	43.18 ± 11.45	57.09 ± 8.84	0.54	6.25 ± 3.18	8476 ± 3365

**Table 2 antibiotics-12-00307-t002:** Tigecycline main pharmacokinetic parameters estimated after the administration of the 7th dose of the standard regimen to 11 patients hospitalized due to an acute COPD exacerbation.

	Patient 1	Patient 2	Patient 3	Patient 4	Patient 5	Patient 6	Patient 7	Patient 8	Patient 9	Patient 10	Patient 11
C_max pl_ (ng/L)	318.45	405.8	640.4	905.6	978.8	1649.2	1679.9	1184.3	444.4	858.8	1670.1
C_min pl_ (ng/L)	10.5	41.2	3.1	159.8	286.18	324.27	358.1	142.99	289.1	378.4	365.65
C_max sp_ (ng/L)	818.2	388.5	410.8	799.6	408.7	897.2	510.8	n/a	427.3	1172.0	586.0
C_min sp_ (ng/L)	293.0	225.3	271.7	248.3	344.2	354.8	322.2	n/a	261.4	450.6	309.1
AUC _0–12 pl_ (ng∙h/L)	645.53	1661.8	1491.7	2841.73	4815.69	4553.50	6399.67	3457.45	4232.27	4717.10	6608.46
AUC _0–12 sp_ (ng∙h/L)	3360.65	3510.8	4046.9	4199.8	3314.77	4528.12	4700.20	n/a	4024.92	5726.05	2820.6
Penetration Ratio	5.20	1.90	2.70	1.40	0.68	0.99	0.70	n/a	0.90	1.20	0.42
T_1/2_ (h)	2.7	7.9	1.6	28.8	23.9	10.3	33.4	12.1	25.5	n/a	25.5
t_max-sp_ (h)	3	1	3	1	3	6	6	n/a	3	2	1

C_max pl_: peak concentration in plasma, C_min pl_: minimum plasma concentration, C_max sp_: maximum sputum concentration, C_min sp_: minimum sputum concentration, AUC _0-12 pl_: area under the curve plasma 0–12 h in plasma, AUC _0–12 sp_: area under the curve in sputum 0–12 h, Penetration ratio: AUC _0–12 sp_/AUC _0–12 pl_. T_1/2_: elimination half-time, t_max-sp_: time to reach c_max_ in sputum, n/a: not applicable.

## Data Availability

Data is contained within the article.

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
