# Peer review of "Tigecycline Pharmacokinetic and Pharmacodynamic Profile in Patients with Chronic Obstructive Pulmonary Disease Exacerbation"

_antibiotics, 2023, doi:10.3390/antibiotics12020307_

Round 1

Reviewer 1 Report

1-    Title: Please revise title … to Chronic Obstructive Pulmonary Disease as this is the standard name of the disease

2-    Make the abstract structured: background, method, results, conclusion

3-    Based on the MDIP author instruction, the section “key points” is not required, so please delete all this section together

4-    Introduction: merge the first and second paragraphs together 

5-    Try to shorten the details in the introduction section 

6-    You have 2 sections that should be one: “2. Study Design, Participants and Procedures” and “3. Methods”. You should have one section “Materials and methods” and Study design, participants, etc should be sub-section of the “Materials and methods”, please revise 

7-    You should add a paragraph in the discussion section about the limitations of your study, also please add that the small sample size of 11 is one of your limitation s

Author Response

Thank you very much for your comments. Changes and responses are point to point explained here:

1-    Title: Please revise title … to Chronic Obstructive Pulmonary Disease as this is the standard name of the disease: the title has been revised (line 3)

2-    Make the abstract structured: background, method, results, conclusion.

The abstract was made divided in the suggested sections (lines 25,27, 30, 35).

3-    Based on the MDIP author instruction, the section “key points” is not required, so please delete all this section together: key points were deleted

4-    Introduction: merge the first and second paragraphs together: 1st and 2nd paragraph of the introduction were merged (line 45)

5-    Try to shorten the details in the introduction section 

Line 51: .  Due to its extended antimicrobial activity, empirical use either as monotherapy or combination therapy was expected since its commercial launch in 2005 for USA and 2006 for Europe. was erased

Line 58: in view of published meta-analyses linking its use with an increased mortality rate by 0.6% and was erased

Line 59 was shortened to: Since then, tigecycline seems to seek its role in antimicrobial regimens for respiratory infections 

6-    You have 2 sections that should be one: “2. Study Design, Participants and Procedures” and “3. Methods”. You should have one section “Materials and methods” and Study design, participants, etc should be sub-section of the “Materials and methods”, please revise: sections have been revised accordingly (lines 80,81, 121-122,136)

7-    You should add a paragraph in the discussion section about the limitations of your study, also please add that the small sample size of 11 is one of your limitation s: a paragraph with the study limitations has been added (lines 327- 332)

There are several limitations to our study. COVID-19 pandemic led to major changes in clinical practice worldwide as well as in both the pulmonary wards where patients eligible for this study were hospitalized and is the main cause for the limited sample size of our study. Furthermore, since this study was designed for non-intubated COPD patients in everyday clinical practice, sputum samples were collected instead of bronchoscopy lavages, which could provide tigecycline ELF concentrations.  

Furthermore, we replaced the figure 1 with a better quality one and some English editing corrections were made in the manuscript. We appreciate your valuable comments and hope you find our manuscript suitable for publication.

Reviewer 2 Report

The article “Tigecycline pharmacokinetic and pharmacodynamic profile in  patients with Chronic Pulmonary Obstructive Disease exacerbation” is on a very interesting topic. I have the following comments/suggestions,

1. The abbreviation pl is used to describe what? Define it in the abstract.

2. The abbreviation sp and Cmax sp are used to describe what? Define it in the abstract.

3. Please give some information on the PK of Tigecycline in the introduction section with particular reference to AUC, T1/2, Vd.

4. The Ethical approval number should be added with the provided Bioethics Committee name.

5. Please provide details for the performed non-compartmental analysis, by which method the AUC was calculated, how many conc vs. time points were included, and how was the T1/2 calculated.

6. Please keep the subscripts uniform (line no 160).

7. It is reported that changes in plasma protein conc (albumin) occur in COPD; the authors should also discuss this point with this study's results in the discussion section. 

Author Response

Thank you very much for your comments and suggestions on our manuscript. Point to point answers and changes in the manuscript are provided below. 

  1. The abbreviation pl is used to describe what? Define it in the abstract.

Line 30: In plasma, mean Cmax pl was…

  1. The abbreviation sp and Cmax sp are used to describe what? Define it in the abstract.

Line 31: In sputum, mean Cmax sp was 641.91 ±253.07 ng/mL and mean Cmin sp was 308.06±61.7 ng/mL

  1. Please give some information on the PK of Tigecycline in the introduction section with particular reference to AUC, T1/2, Vd.

Details on tigecycline PK are given in the Discussion section (lines 224-228): Tigecycline pharmacokinetics remain controversial with many unanswered questions, mainly risen from the problematic clinical efficacy in severe lower respiratory illness, such as VAP and HAP [22,23,24]. In view of the increased mortality and microbiological failure [25], high dosage regimens of tigecycline have been proposed in order to overcome the probable pharmacokinetic basis of reduced efficacy.

  1. The Ethical approval number should be added with the provided Bioethics Committee name.

Ethical approval number has been added (line 349): Institutional Review Board Statement: The study was conducted in accordance with the Declaration of Helsinki and approved by the Aristotle University Medical School Bioethics Committee (No:154/2015) and lines 87-88: The study was approved by the Aristotle University Medical School Bioethics Committee (No:154/2015).

  1. Please provide details for the performed non-compartmental analysis, by which method the AUC was calculated, how many conc vs. time points were included, and how was the T1/2 calculated.

Details about time blood and sputum time points are provided in lines 109-122. A loading dose of 100 mg tigecycline (dose 1) was administered intravenously (i.v) to all patients followed by 50 mg tigecycline i.v every 12 hours. Every tigecycline infusion duration was 1 hour. Blood and sputum samples were collected at steady state, after tigecycline dose 7 (on day 4 or 5 of hospitalization). Blood samples were collected through a peripheral catheter positioned for this purpose remotely from the catheter used for the treatment infusion, on specific time points i.e., just before the infusion of tigecycline and 12 hours from the previous dose (time 0), on completion of the 1-hour tigecycline infusion (time 1) and then 2,3,4,6,9 and 12 hours after the iv administration. Blood samples were centrifuged, and the supernatant plasma was stored at -20oC.

Mean plasma and sputum concentrations are plotted against time in figure 1.  

AUC, Vd and T1/2 were calculated automatically by the Win-Nonlin software, v.8.3, by a non-compartment model (lines 153-155).

  1. Please keep the subscripts uniform (line no 160).

(previous) Line 160- (now line 158) has been changed accordingly: by the FDA MIC breakpoints for S. pneumoniae...

  1. It is reported that changes in plasma protein conc (albumin) occur in COPD; the authors should also discuss this point with this study's results in the discussion section.

Line 73 was changed to: COPD patients are more likely older, active or ex-smokers, underweight with low albumin levels, suffering from various comorbidities, and on various inhaled and systemic medication.

Thank you for your suggestion. We intended to discuss so in our study, however our 11 moderate COPD patients were of a high median BMI (table 1) and none of them had low protein or albumin levels. Consequently, we could not relate our findings with hypoalbuminemia in this study.     

Furthermore, we replaced the figure 1 with a better quality one and some English editing corrections were made in the manuscript. We appreciate your valuable comments and hope you find our manuscript suitable for publication.

Round 2

Reviewer 2 Report

The authors have addressed all of my comments in their revised submission.